# Can knowledge reclassification accelerate technological innovation?

Peter Persoon[1,2¤]*

1 Oxford Martin School, University of Oxford, Oxford, United Kingdom, 2 Animal Sciences, Wageningen University and Research, Wageningen, The Netherlands

¤ Current address: Wageningen University and Research, Droevendaalsesteeg 4, 6708 PB Wageningen, The Netherlands

* peter.persoon@wur.nl

## Abstract

Technological knowledge evolves not only through the generation of new ideas, but also through the reinterpretation of existing ones. Reinterpretations lead to changes in the classification of knowledge, that is, reclassification. This study investigates the relationship between reclassification and the rate of technological change both empirically and analytically. Drawing on patent data as a proxy for technological knowledge, I discuss two empirical patterns: (i) more recent patents are more likely to get reclassified and (ii) larger technological classes acquire proportionally more reclassified patents. These patterns are then used to formulate an analytical model of knowledge accumulation, which can be solved exactly to describe the relationship between reclassification and growth rates in detail and to make further predictions about related knowledge quantities. These predictions are supported across all major technology domains, implying a strong link between reclassification and the pace of technological advancement. In sum, the model allows for a better understanding of both the magnitude and shape of technological growth patterns. More generally, it connects various seemingly unrelated knowledge quantities, providing a basis for knowledge-intrinsic explanations of growth patterns.

## 1. Introduction

Technology and science advance as new ideas are added to a cumulative structure of earlier ideas. Equally important, however, are the developments that restructure or reorganize earlier ideas, which occur for example when technologies find new applications or when scientific results are reinterpreted. Think about how solar panels, once used primarily in satellites, are now installed on rooftops around the world. Or think about how Einstein's work on the photoelectric effect partly revived Newton's corpuscular theory of light. While these changes may not directly introduce new words or inventions, they potentially alter the categorization or classification of existing knowledge, i.e., they reclassify existing knowledge. Where the societal relevance

**Data availability statement:** The raw data underlying the results presented in the study are available through purchase at the European Patent Office, https://www.epo.org/en/searching-for-patents/business/patstat. The patent office will need to be contacted for the older versions of Patstat. A processed dataset is available at https://doi.org/10.17026/SS/MEV4KE.

**Funding:** The author(s) received no specific funding for this work.

**Competing interests:** The authors have declared that no competing interests exist.

of generating new ideas is undisputed, the direct or indirect relevance of reclassification is less clear. This is mainly because it is poorly understood how substantial the effect is and what the underlying dynamics are.

However, studying the dynamics of reclassification is not trivial as it mixes two processes for a given class of knowledge: first, the inclusion of new ideas in a cumulative process of creation; and second, the inclusion/exclusion of older ideas as a result of ongoing reclassification; see Fig 1. It is likely that the two types strongly interact, but how exactly largely remains an open problem. Earlier contributions have focused on either the process of new idea generation [1–4], or the process of classification and reclassification [5–8]. In this contribution, I aim to integrate these dynamics in a single analytical model and subsequently derive how reclassification and growth relate to one another. Subsequently, I check the model predictions in an empirical analysis where patent classification data are used as a proxy for technological knowledge. A detailed analysis of the dynamics of reclassification advances our understanding of knowledge creation on various fronts.

First, it provides a clue as to where the large variation in growth and improvement rates across different technologies comes from [9,10]. Given the close relationship between technological development, economic growth, and societal change, this continues to be one of the central issues in the innovation literature [11,12]. Some have pointed to conceptual links between technologies as predictors of innovation rates [13,14], while others have emphasized structural features such as technological complexity [9,15]. Complementing this literature, this contribution finds that technological classes that are often reclassified tend to grow faster. Although further research is needed to explore causality and the exact mechanism underlying this pattern, the results suggest there is a connection between a sensitivity to redefinition – possibly rooted in a wider scope or applicability of a technology – and the rate at which technological ideas multiply. This provides a relevant perspective on the growth rates of General Purpose Technologies, which by definition adhere to a wide range of applications [16,17], and more generally sheds new light on the debate about the cumulativeness of technological knowledge [18–20], i.e., the extent to which technologies build on earlier results.

Second, the findings in this research have implications to the sociology and philosophy of science and technology. In theories of 'paradigm shifts', reinterpretations are an indispensable part of scientific [21] and technological [22] revolutions. This contribution offers a quantitative, empirical perspective on reinterpretation and more generally reclassification, confirming these processes play a key role in technological developments. However, where in theories of paradigm change, reinterpretation is often presented as a short, revolutionizing event, in this contribution it is instead implemented as a routine, ongoing process.

Third, explicitly incorporating reclassification into models of technological growth provides a fresh explanation for the commonly observed "drop" in recent patent counts—often referred to as a truncation effect [23,24]. This phenomenon is not fully explained by administrative factors, such as the 18-month lag between patent filing and publication in the U.S. Especially when it is focused on a particular technology

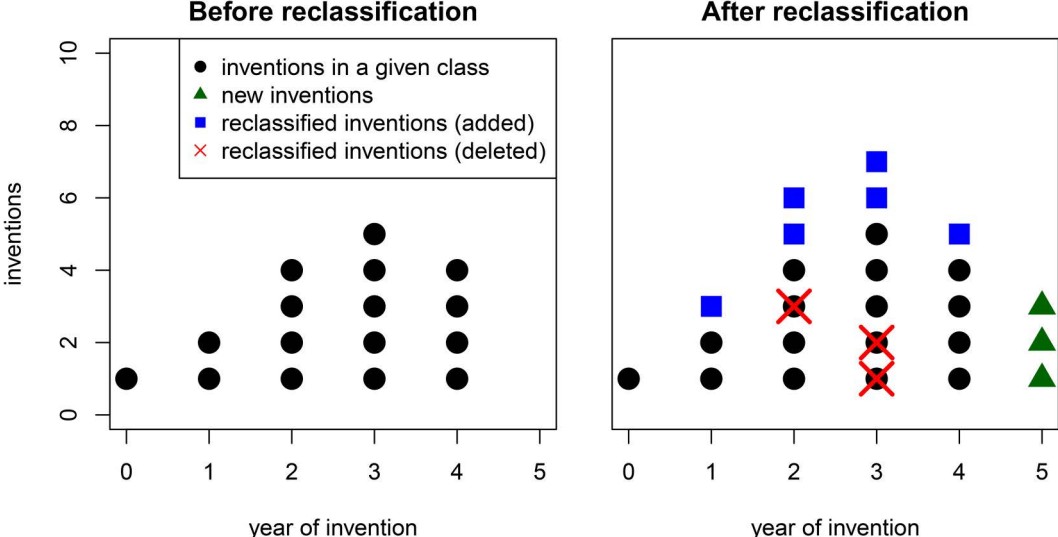

**Fig 1. A mix of two processes.** The time development of a class of knowledge mixes two effects: the introduction of new inventions each year (triangles) and the inclusion/exclusion of older inventions as a result of reclassification (squares/crosses).

such as green technology, there is a risk in interpreting this drop as a real decline in patenting rates [25–28], even though several years later the drop seems to have disappeared [29]. This research shows that such declines can be a natural consequence of reclassification patterns and predicts the timing of these drops with high accuracy.

The paper proceeds in three parts. The first part is phenomenological, systematically observing patent reclassifications and discussing two empirical patterns. The second part is analytical, where the empirical patterns are used to develop a model relating reclassification to growth. The third part tests the model's predictions against a range of empirical outcomes.

## 2. Phenomenology of reclassification

Patent classifications are expert-based categorizations of inventions and therefore provide an ideal empirical definition of individual technologies. There are also limitations to using patent data: not all technologies are patented and many patented inventions eventually find no practical application. In addition, patent classifications are primarily designed and regularly updated to assist patent examiners in their search for prior art. Despite these limitations, it is difficult to think of an alternative source of codified technological knowledge that could match the scope and detail of patent data. Patent classification systems are continuously updated by adding (or deleting) classes and classifications of patents in a process called 'reclassification', thereby accommodating changing societal perspectives and technological developments that do not fit the existing scheme [5, 7]. Take for example US patent # 4671271 from 1987, which describes a type of protective facial mask: where it was initially only classified as a protective garment (CPC group A41D13), in a classification version of 2016, it is also classified as device for influencing the respiratory system (CPC group A61M16). This is therefore an example of a patent obtaining extra classifications, but also, an example where a group in the classification system obtains an extra patent (even though it is an old patent). There may be several reasons why this happened: perhaps a patent examiner familiar with the group came across the patent by chance, perhaps the definition of the group was modified such that it now includes this patent, or perhaps the classification algorithm was improved. Despite regular publications of patent offices about changes to the classification system, there is generally no clear-cut way to find the reason for individual cases of reclassification. However, by keeping old versions of the patent classification system and comparing it to newer systems, one can systematically study the general patterns of reclassification.

Previous studies have pointed out several regularities in the patent (re)classifications. Research on US patent classifications showed that, since the start of the 20th century, most patents have a combination of more than one classification, that the number of distinct classes increases as a power law with the number of patents, and that the number of used combinations increases proportionally with the number of patents [6,7]. Given these long-term dynamics, let me focus instead on the changes happening within a single reclassification moment. To count each invention only once, the unit of analysis in this research is a patent family. This is a collection of patent documents that all correspond to the same invention (but are filed, for example, at different patent offices). To be precise, I will work with docdb families as administered by the European Patent Office, in which all documents have the same priority patent application, i.e., the earliest application for a patent with this invention). I will consider patent families filed between 1970 and 2019, each including at least one US application and one filed in another jurisdiction. This comprises more than 5 million families, in the following also referred to simply as 'patents'. Using data from 4 Patstat editions (2013, 2016, 2019, and 2023), there are three distinct reclassification moments: 2013/2016, 2016/2019 and 2019/2023. Reclassification happens yearly, so I aggregate the changes over approximately three years for each reclassification moment. I will use the Cooperative Patent Classification (CPC) on three levels: the section level (1 digit), subclass level (four digits), and main group level (four digits plus main group number) [30].

For a given reclassification moment and any technological class, I consider both the number of families added to that class ('positive reclassifications') and those deleted ('negative reclassifications'). The positive and negative reclassifications from each CPC subclass are plotted for the size of those subclasses (in number of families) in Fig 2. This plot focuses on the reclassification moment 2019/2023. It is almost the same for the other reclassification moments, see also the appendix in [29] for a similar plot on CPC class level. The plot only considers families that were already there in 2019,

**Subclass size and number of reclassifications 19/23**

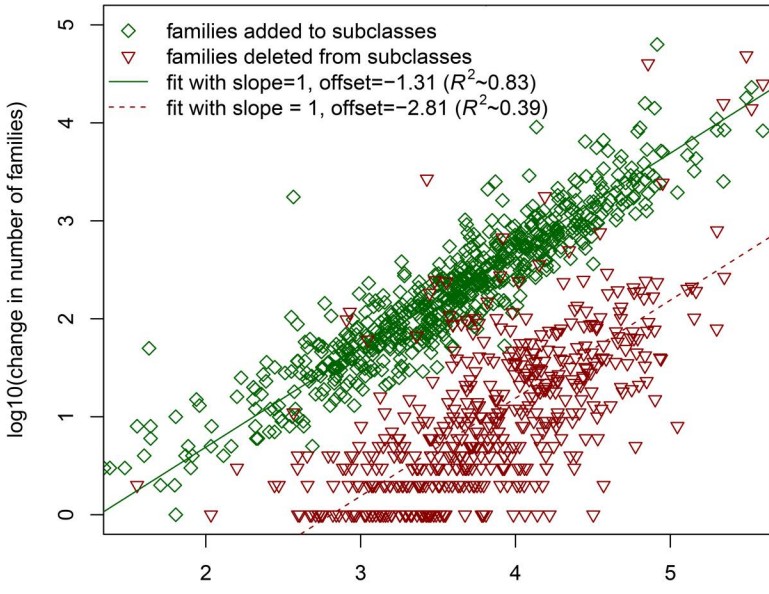

Legend:
- ◇ families added to subclasses
- ▽ families deleted from subclasses
- —— fit with slope=1, offset=−1.31 ($R^2$~0.83)
- ---- fit with slope = 1, offset=−2.81 ($R^2$~0.39)

y-axis: log10(change in number of families)
x-axis: log10(size of subclass in number of families)

**Fig 2. Reclassifications are proportional to (sub)class size.** This figure shows the number of families added and deleted for all subclasses for the reclassification moment 2019/2023. Note that the axes are log-transformed, hence the fits with slope 1.0 suggest the number of reclassifications is proportional to subclass size. The offset (the y-coordinate of the fit intersection with the y-axis) that maximizes $R^2$ for the positive reclassifications is −1.31. Plotting the fit without transforming the axes thus results in a linear relation with a slope of $10^{-1.31} \approx 0.05$.

thus making sure that any patent added is necessarily a reclassification and not a new patent. This plot shows that both the positive and negative reclassifications of a class are proportional to the size of that class. For the negative reclassifications, this is intuitive: if each classification is equally likely to be deleted, the larger classes lose proportionally more families. For the positive reclassifications, however, this is more puzzling: why would larger classes attract more reclassifications? One possible explanation is that the scope of a technological class somewhat widens with each new patent, and with a wider scope, more families can be classified into that class. The pattern in Fig 2 suggests that the positive reclassifications are generally more substantial than the negative reclassifications. This is confirmed by an additional analysis in the Supplementary Information (SI), where the net reclassifications are considered on the subclass level and where the role of negative reclassification is investigated in more detail.

Next, I obtain the net reclassifications for each filing year by subtracting the total negative reclassifications from the total positive reclassifications of families with that filing year. In Fig 3 I plot the net reclassifications over the total number of classifications for each filing year, each panel corresponding to a different reclassification moment. The plot shows more reclassifications per classification for patents with filing years close to the reclassification moment, in other words, recent patents are more likely to get reclassified. The pattern is similar across the three reclassification moments, except that for the 2013/2016 moment, where the net reclassifications are negative for some years. The reason for this is unclear but might relate to the fact that the CPC was still young in those years and parts were still being reorganized. A log-log plot of the positive and negative reclassifications, see S2 Appendix, indicates an inversely proportional relation between the number of reclassifications per classification $r$ and the time between cohort filing year $c$ and time of reclassification $t$, i.e., $r \propto \frac{1}{t-c}$. Fig 3 includes fits of the data based on this relation, introducing the 'net reclassification rate' $\beta$. Because each reclassification moment in the data aggregates three years, I add $r(c, t) = \frac{\beta}{t-c}$ for three subsequent values of $t$ to construct these fits, see S2 Appendix. A large, positive $\beta$ indicates more classifications are being added, while a negative $\beta$ suggests classifications are being deleted. While this relation fits the data rather well, I observe substantial variation across the three moments in $\beta$. Especially in the 2019/2023 reclassification moment, many extra classifications were added to the system. The values for $\beta$ appear to vary across different CPC (sub)classes too: some receive relatively more

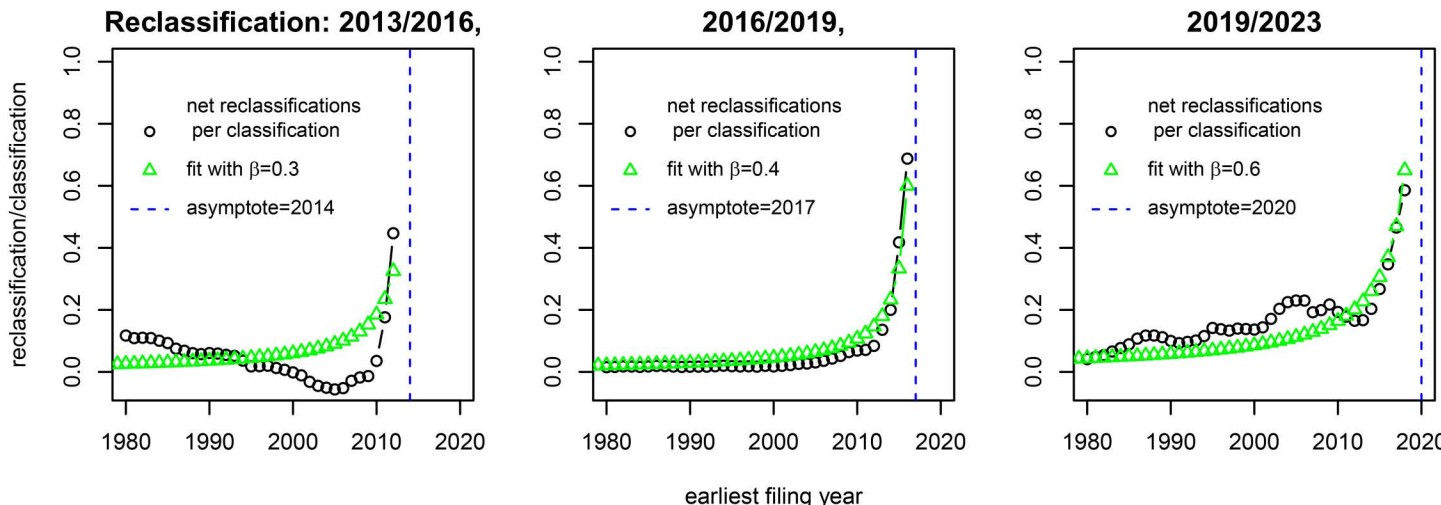

**Fig 3. Recent patents are reclassified more.** For three reclassification moments (2013/2016, 2016/2019, and 2019/2023), I calculate the net reclassifications and divide this by the total number of classifications prior to reclassification. I plot this fraction for the earliest filing year of the corresponding patent families. As the filing years c get close to the moment in time of reclassification t, the net reclassifications per classification r(c,t) sharply increase. I include fits based on the relation $r(c, t) = \frac{\beta}{t-c}$ for different values of $\beta$. For this plot, the classifications are aggregated on subclass level. Only the families occurring in the classifications systems before and after reclassification are included.

reclassified families than others. This leads to the question whether some technologies are more prone to reclassification than others, i.e., having intrinsically higher $\beta$s. Again reasoning from the scope or applicability of a technological class, possible candidates for greater $\beta$ could be general (or multi) purpose technologies [16,17]: wide scopes may be more likely to attract reclassifications from other classes, (thereby widening the scope again).However, measuring multi-applicability in patents is challenging and even if that could be done, then fitting and estimating $\beta$ for smaller classes also brings in additional challenges. I therefore leave investigating the variability of $\beta$ for later work.

That recent patents are more likely to get reclassified makes sense for several reasons. First, when new classes are introduced, they likely consist mostly of recent patents: if they consist of mostly older patents, it would be puzzling why the class was not introduced earlier. Second, recent patents may not yet be well understood and new applications are still being discovered. Third, the language of recent patents may be more modern and the classification (search) algorithms still need some time to adjust to this language.

## 3. A model of reclassification and growth

In this section, I will explore analytically how reclassification and growth could interact. I will introduce a model that describes the growth of a technology over time incorporating two effects: (i) how inventions in that technology trigger new inventions and (ii) how that technology is reclassified so that it includes inventions from other technologies. Let me denote the number of patents in that technology with filing year $c$ by $n_c$: this group of patents will be referred to as a 'cohort'. Generally, the number of patents in a cohort will change over time: with later reclassifications, new patents might be added and subtracted. I am therefore interested in describing how $n_c$ depends on time $t$, i.e., $n_c(t)$. To incorporate (i), I use the principle that knowledge develops cumulatively, i.e., that new knowledge builds on old knowledge. [18–20] This can be implemented by allowing any earlier result in a technology to at any time trigger a new result in that technology. The triggering of new patents, essentially creating the new cohort $c$, can therefore be written as $\Delta_t n_c(t) \propto n(t)$, where $n(t) = \sum_c n_c(t)$. Note however that this only counts for what will be the new cohort at time $t+1$, i.e., only for $c = t+1$. To incorporate (ii), I explicitly use the empirical patterns identified in Section 2. The first pattern suggests that the probability for a patent from cohort $c$ to get reclassified at time $t$ is inversely proportional to $t-c$. The second pattern suggests that the greater the class, the more families are reclassified into or out of that class. I will focus on the net-reclassification effect, hence subtracting the negative reclassification from the positive ones. The change in $n_c(t)$ as a result of reclassification at time $t$ can then be written as $\Delta_t n_c(t) \propto n_c(t)/(t-c)$. Note that these changes apply to all cohorts except the newly introduced cohort, so for this expression, $c$ will be smaller than $t+1$, i.e., $c < t+1$. Finally, one cannot reclassify or trigger patents into future cohorts, hence $\Delta_t n_c(t) = 0$ for $c > t+1$. The three dynamical relations and corresponding cohorts can therefore be summarized in the following set of equations:

$$\Delta_t n_c(t) = \begin{cases} 0 & \text{if } c > t+1, \\ \alpha n(t) & \text{if } c = t+1, \\ \beta n_c(t)/(t-c+1) & \text{if } c < t+1, \end{cases}$$

(1)

where I introduce the technology-dependent triggering rate $\alpha \geq 0$ and net reclassification rate $\beta$. In the following, I will focus on the case where $\beta > 0$ but the model in principle also allows for negative reclassification rates. In the S1 Appendix I discuss in more detail what happens if $\beta < 0$.

Let me focus on the simplest case, a technology with one patent at $t = c = 0$, i.e., $n_c(0) = 1$ for $c = 0$ and $n_c(0) = 0$ otherwise. As I show in more detail in the S1 Appendix, the third line of Equation 1 directly leads to the relation

$$n_c(t) = \binom{t-c+\beta}{\beta} n_c(c).$$

(2)

which can be used to derive the exact solution

$$n_c(t) = \binom{t - c + \beta}{\beta} \sum_{u=0}^{c} \binom{u\beta + c - 1}{c - u} \alpha^u.$$

(3)

Summing over $c \geq 0$ yields the total number of families

$$n(t) = \sum_{u=0}^{t} \binom{(u + 1)\beta + t}{t - u} \alpha^u.$$

(4)

In Fig 4, I plot $n_c(t)$ for $c$, three values of $t$ and fixed $\alpha = 0.05$ and $\beta = 0.7$. It is clear that for large $t$, the number of patents grows exponentially (note the y-axis is logarithmic) before their numbers drop again in years close to $t$. To better understand how the base of the exponential relation depends on $\alpha$ and $\beta$, it is instructive to calculate the generating function $G(z)$ of $n(t)$, that is, $G(z) = \sum_{t=0}^{\infty} n(t)z^t$ for some $0 \leq z < 1$. In the S1 Appendix, I demonstrate that this evaluates to

$$G(z) = \frac{1}{(1 - z)^{1+\beta} - \alpha z}.$$

(5)

Furthermore, following a theorem from generating functions theory ([31], see page 341), I show that this rational generating function allows for an asymptotic estimate of $n(t)$. For real $\beta > 0$ and $0 < \alpha < 1$, it counts

$$n(t) \simeq n_0 g^t,$$

(6)

where $n_0$ is a constant depending on $\alpha$, $\beta$, and the initial conditions. The base of the exponential relation $g$ is a real number satisfying

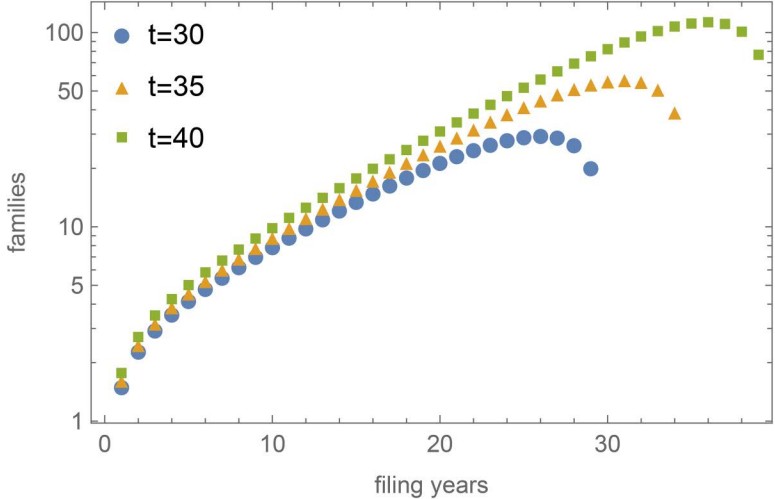

**Fig 4. Model solutions plotted.** Following the solution in Equation 3, this figure shows the number of patents in each cohort $n_c(t)$ plotted for each filing year $c$ and for three values of time $t = 30, 35, 40$. I choose the values $\alpha = 0.05$ and $\beta = 0.7$. The number of patents increases exponentially until the apparent 'decline' when the filing year $c$ of a cohort is close to $t$, that is, for recent cohorts.

$$\left(1 - \frac{1}{g}\right)^{1+\beta} = \frac{\alpha}{g}. \tag{7}$$

While this implies one cannot obtain a closed expression of $g$ in terms of $\alpha$ and $\beta$, Equation 7 is enough to get a qualitative understanding of the effect of reclassification on growth. As I show in the appendix, $g$ is always larger for larger $\alpha$ and larger $\beta$ and $1 + \alpha \le g < 1 + \alpha + \beta$. When both $g - 1 \ll 1$ and $\alpha \ll 1$, i.e., for slow growth, $g \approx 1 + \alpha^{\frac{1}{1+\beta}}$. The model therefore suggests that, in the regime of slow technological growth, there is a positive, non-linear relation between the rate of reclassification and the rate of growth. Note that the upper bound $g < 1 + \alpha + \beta$ implies that in a regime of fast technological growth, the relation between the rate of reclassification and growth rates is linear instead.

Using the model solutions, I can calculate several interesting quantities that allow for empirical validation of the model. In the following, I discuss how to interpret and calculate three of these quantities: the decline time, reclassification proportion and number of classifications per patent.

### 3.1. Decline time

Using the model solutions, I can calculate several interesting quantities that allow for empirical validation of the model in the next section. Let me first consider the year for which the number of patents peaks before they (seemingly) decline in recent years, again see Fig 4 for an illustration. As is clear from this figure, the peaks are not stationary but instead shift to more recent years as $t$ increases: the decline is therefore only apparent. Let me study the typical 'decline-time' $t_d = t - c$ between the filing year $c$ where the peak is and the present time $t$. If the peak occurs at filing year $c$, then that is the earliest year such that $n_c(t) \ge n_{c+1}(t)$, or, using Equation 2,

$$\binom{t - c + \beta}{\beta} n_c(c) \ge \binom{t - c + \beta - 1}{\beta} n_{c+1}(c + 1). \tag{8}$$

From the initial conditions and the second line of Equation 1, note that $n_{c+1}(c + 1) = \alpha n(c)$. Approximating $n_{c+1}(c + 1) = \alpha n_0 g^c$, substituting in Equation 8 and simplifying then gives

$$\frac{\beta}{t - c} \ge g - 1 \tag{9}$$

The $c$ for which the left- and right-hand sides are equal is the peak year, hence the typical time-span between peak and present can be approximated by

$$t_d \approx \frac{\beta}{g - 1}. \tag{10}$$

### 3.2. Reclassification proportion

Second, I consider the total number of reclassified families in year $t$, denoted by $v(t)$. The phenomenology in Section 2 requires that $v(t)$ is proportional to the total number of families. Let me show the proportionality is indeed predicted by the model and at the same time derive an explicit expression for the constant of proportionality $V$, which I will refer to as the 'reclassification proportion'. Equation 1 leads to $v(t) = \sum_{c=0}^{t-1} \beta \frac{n_c(t)}{t-c}$. Rather than doing the summation, let me calculate the total number of families added in year $t$ by summing Equation 1 over all $c$, obtaining

$$\sum_{c=0}^{t} \Delta_t n_c(t) = \alpha n(t) + \beta \sum_{c=0}^{t-1} \frac{n_c(t)}{t-c}$$

(11)

$$\Delta_t n(t) = \alpha n(t) + v(t).$$

(12)

Dividing left and right by $n(t)$, using the approximation for $n(t)$ in Equation 6 and summing over $t$, this becomes

$$(g-1)t \simeq \alpha t + \sum_{t'=1}^{t} \frac{v(t')}{n(t')}.$$

(13)

For the left- and right-hand side to agree for large $t$, I conclude that $v(t) \simeq n(t)(g-1-\alpha)$. As required, the number of reclassified families is proportional to the total number of families; furthermore, the proportion of reclassification is expected to be $V = g-1-\alpha$.

### 3.3. Number of classification per patent

Finally, the current formulation of the model is focused on the development of one individual technological class, where $n_c(t)$ represents a set of unique patents with filing year $c$ at time $t$. Note that if this set would be the collection of all patents, i.e., 'all technology', then $\beta = 0$ because there are no external classes or patents to draw from. However, approaching this slightly differently, the model can still be sensibly applied to the union of all technologies: if $n_c(t)$ instead represents all unique classifications of patents with filing year $c$ at time $t$ (that is, counting patents with multiple classifications multiple times), then $n_c(t)$ does increase for $c < t$. In this interpretation, there is a relevant quantity $W_0(t)$ describing on average how many classifications each new patent has (upon introduction). Assuming $W_0$ is constant (or changes very slowly), the number of unique patents is then given by $\sum_c n_c(c)/W_0$, allowing us to calculate the number of classifications per patent, denoted by $W$ (for more details see S1 Appendix),

$$W = \frac{n(t)}{\sum_{c=0}^{t} n_c(c)/W_0},$$

(14)

$$\approx W_0 \frac{g-1}{\alpha}.$$

(15)

Another interpretation of this expression is that the growth rate $g-1$ is proportional to $W$ (the proportionality constant being $\frac{\alpha}{W_0}$): in other words, technologies or technological classes with more classifications per patent are expected to grow proportionally faster.

## 4. Model validations

In this section, I explicitly check the main predictions made by the model using patent classification data. More specifically, based on the estimated values for the model parameters $\alpha$ and $\beta$, I will compare the predicted values with the empirical values for four interesting quantities.

1. the overall growth factor $g$,

2. the decline-time $t_d$,

3. the reclassification proportion $V$,

4. the number of classifications per patent $W$.

Let me apply the model to 'all technology' (see Section 3.3). Based on the definitions in Equation 1, patent numbers between 2010–2015, and the fits in Fig 3, the values for $\alpha$ and $\beta$ can be directly estimated (for a detailed explanation, see the S2 Appendix). I find $0.024 < \alpha < 0.027$ and $\beta \approx 0.4$.

As a first validation, using $0.024 < \alpha < 0.027$ and $\beta = 0.4$ in Equation 7, I predict an overall growth-factor of $1.07 < g < 1.08$. Measuring $g$ using OLS fits for the log number of classifications between 1980 and 2015 using the 2023 dataset gives $g \approx 1.079$, see Fig 5. For the other datasets and using the cumulative number of classifications instead, I find similar values for $g$ between 1.075 and 1.08. All of these are very well fitted by an exponential relation ($R^2 \approx 0.98$). I conclude that Equation 7 agrees rather well with observation.

As a second validation, I use $g = 1.079$ and $\beta = 0.4$ in Equation 10 to predict the decline-time $t_d \approx 5.1$. Fig 5 plots the total number of classifications of all patents by their earliest filing year and clearly demonstrates the apparent declines in recent years. Note also the similarity with Fig 4. A closer examination points out that classifications in the 2013, 2016, 2019, and 2023 data peak respectively in 2006, 2012, 2013, and 2014. This leads to decline-times in these datasets of 7,4,6 and 9 years. These values are close to the predicted value for $t_d$, the only exception being 2023. However, if I choose $\beta = 0.6$ for the 2023 data (see Fig 3), the prediction is that $t_d \approx 7.6$. To demonstrate that the decline-time is also observed for each CPC section and a number of subclasses, I include plots of their time development for various versions of Patstat in S2 Appendix. For those cases too, I observe decline-times between 3 and 6 years.

As a third validation, I use the estimated (lower) values for $g$ and $\alpha$ in the expression $V = g - 1 - \alpha$ to predict a higher estimate for the reclassification proportion $V \approx 0.056$. The easiest way to measure this value is to determine, for a given reclassification moment, the total net reclassifications and divide by the number of classifications before reclassification.

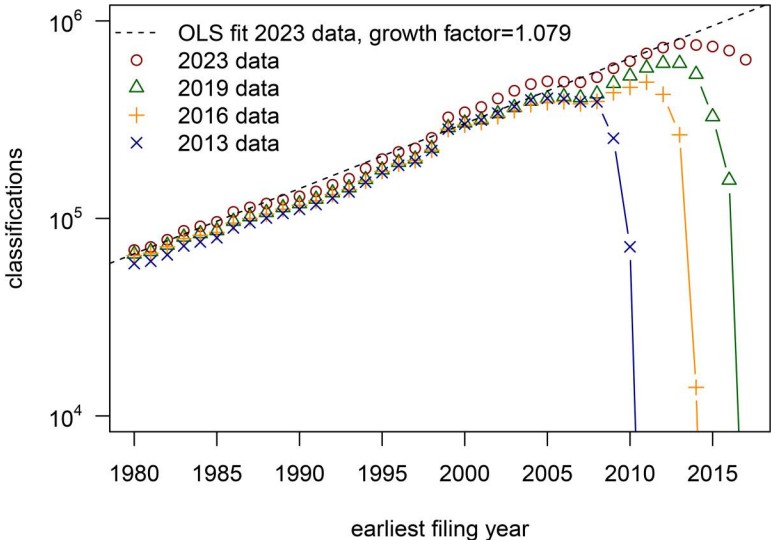

**Fig 5. Total classifications over the years.** This figure shows the total number of classifications of patents by earliest filing year according to 4 different datasets. I include a fit of the 2023 data ($R^2 = 0.98$), which indicates that classifications grow exponentially (the y-axis is logarithmic). Note the similarity with Fig 4, especially with the apparent declines in recent years. A closer examination points out that the 2013, 2016, 2019, and 2023 data peak respectively in 2006, 2012, 2013, and 2014.

Using reclassification moments 2016/2019 and 2019/2023, this leads to a reclassification proportions 0.023 and 0.053 respectively. Those values are reasonably close to the predicted higher estimate for $V$. Furthermore, as explained in Fig 2, the slope of the linear relation between the subclass reclassifications and size is of the order 0.05. While this value takes into account three reclassification years, the order of magnitude is in agreement with the predicted $V$. This suggests that the model applies reasonably well also to individual subclasses, that is, to individual technologies.

For the fourth and final validation, I will consider the classifications per patent first for all technology and then specifically for different technologies. To predict this quantity for all technology, I will use Equation 14 and the measured values for $g - 1 \approx 0.079$, $0.024 < \alpha < 0.027$, $W_0 = 1.25$ for the CPC group level (see S2 Appendix) to derive that $3.7 < W < 4.1$ classification/patent. Using $W_0 = 0.7$ for the CPC subclass level, this is $2.0 < W < 2.13$ instead. Measuring the average number of classifications per patent (using the 2023 data) gives 3.6 classifications per patent on the CPC group level and gives 2.14 classification/patent on the CPC subclass level. The measured values are therefore very close to the predicted intervals. Next, I will explicitly calculate the number of group classifications per patent $w_k$ for each CPC group $k$ (considering classifications on the CPC group level). Equation 14 predicts that this is proportional to the growth rate $g_k - 1$ of these groups. Testing if variation across the groups in $w_k$ correlates with variation in $g_k$ is therefore an indirect test of the effect of reclassification. I sub-select all groups with at least one patent each year between 1970 and 2015 and determine the growth factor $g_k$ using an OLS estimation on the log-transformed number of patents each year. The number of classifications per patent $w_k$ is determined by dividing the total number of group classifications of each patent in a group by the number of unique patents in that group. Each group belongs to exactly one CPC section. In Fig 6, I plot each group's $w_k$ and $g_k - 1$ separately for each section. For each section, I observe highly significant and moderately strong correlations between these quantities (see the values for the Pearson correlation coefficient $R^2$). The only exception is Chemistry/

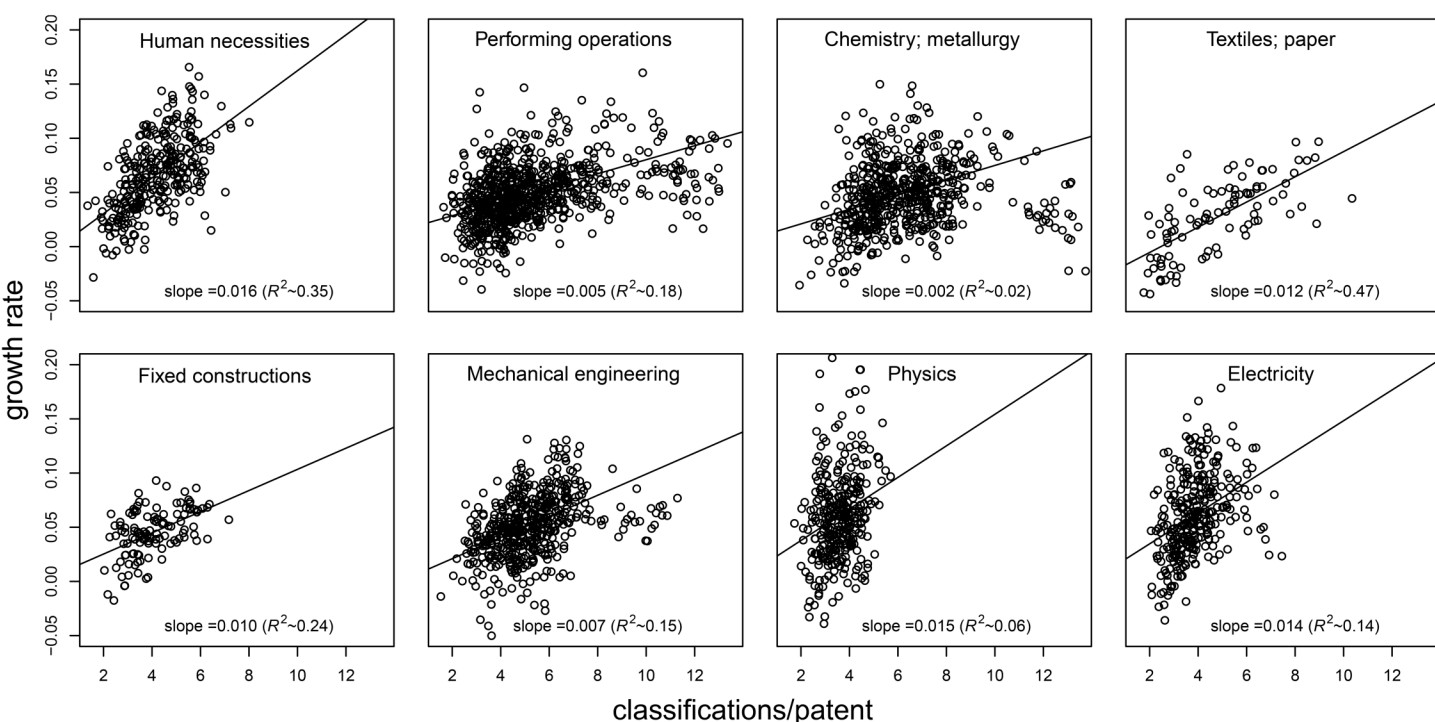

**Fig 6. Growth rates and classifications per patent.** For each CPC group k, I plot the growth rate $g_k - 1$ for the average classifications per patent $w_k$. I plot the groups for each CPC section separately (each group belongs to one section). As predicted, most slopes are of the same order of magnitude as the estimated $\alpha \approx 0.023$.

metallurgy, however, as I show in more detail in the S3 Appendix, the correlations become much stronger when groups with $w_k > 9$ classification/patent (i.e., very high $w_k$) are excluded. Those groups belong to 4 subclasses only, a very small minority. In any case, the relation between $g_k - 1$ and $w_k$ appears to be especially strong for $w_k < 9$. The relations are still highly significant if I control for the total number of patents in each group and if I control for the number of patents in recent years. This indicates that the correlations are not an indirect result of group size or having many recent patents. In the S3 Appendix, I include the details of two more robustness checks. In the first, I reformulate $w_k$ to account for the bias of recent years in reclassifications. In the second, I reformulate $g_k$ to exclude double counting of classifications using a fractional attribution. Even with these reformulations, I observe a strong relation between classifications and growth. Finally, it is interesting to note that the slopes of the observed linear relations, which the model predicts correspond to $\alpha/W_0$, appear to vary across different sections: the slopes are large for Human Necessities, Physics, and Electricity and small for Performing Operations and Chemistry. It is not directly clear where these technology-dependent variations come from. In line with prediction, note that the slopes are of the same order of magnitude as the measured $\alpha$.

## 5. Discussion

This research develops an analytical model relating technological reclassification to the rate of development. Empirical tests using patent classification data are found to be consistent with the model predictions. Possibly, this means that further improvements to categorization and classification could speed up technological innovation, and that reclassified ideas can be considered renewed sources of future ideas. There are, however, also important limitations to this research that I should mention.

First, it is important to stress that the observed positive relationship between reclassification and growth rates is no direct evidence of causality. To explore this further, it is essential to better understand the mechanism of cumulative knowledge development proposed in the model, where earlier patents in a given class 'trigger' the development of later patents within the same class. Reclassification could then accelerate the process, as reclassified patents would trigger new results in multiple classes. Although the 'triggering event' is difficult to observe, earlier work finds a positive relationship between reclassification and increased citation rates [7], suggesting that reclassified patents may indeed have a greater triggering potential. However, the extent to which (patent) citations can be interpreted as knowledge flows is still a matter of debate [32,33]. Of course, another possibility is that there is only indirect causality or causality through other (unobserved) factors, in which case another model would apply.

Second, one should keep in mind the various facets of reclassification: some reclassifications reflect deep conceptual shifts and reinterpretations, while others merely facilitate searching within the patent system and involve less substantial conceptual change. Although the distinction is relevant to the implications of this research, it is difficult to separate these facets empirically.

Third, I should mention that, although the CPC classification is by now widely adopted, there are also other patent classifications systems in use across various countries, most notably the International Patent Classification (IPC). While validating these results with the IPC would be valuable, the CPC is largely derived from the IPC and offers more detailed classifications in many respects.

Finally, I explore the broader implications of this research. This research indicates that reinterpretation, here considered a part of reclassification, is closely linked to the speed of development. This finding aligns well with theories of paradigm change (e.g., [21,22]), which propose that innovation accelerates during paradigm shifts, with reinterpretation playing a central role. In contrast, this study treats reclassification as a continuous process, occurring at a relatively constant rate throughout a technology's evolution and as such allowing for verifiable predictions. This raises the question: should reinterpretation be viewed as a singular event or as a continuous, everyday process?

The results also have some direct implications for research on technological change using patent classifications. The proposed model, based on only two parameters, can explain various seemingly unrelated quantities, such as growth

rates, decline-times, and the number of classifications per patent. It is intuitive to think that these quantities are determined mainly by external factors, such as fluctuations in investments or resource price dynamics. For example the earlier mentioned green decline, which ran over several years, has been attributed to factors such as falling fossil fuel prices and the financial crisis (for an overview see [29]). Yet, as reclassification (i.e., $\beta$) for green technology is relatively strong [29], Equation 10 directly suggests that the decline-time for green technology is relatively large, offering an alternative explanation for the years of green decline. This demonstrates that it is not always necessary to bring in external factors; instead knowledge-intrinsic factors might provide a simpler explanation.

## Supporting information

**S1 Appendix. Analytical model solutions.**
(PDF)

**S2 Appendix. Empirical patterns.**
(PDF)

**S3 Appendix. Robustness checks and OLS results.**
(PDF)

## Acknowledgments

The author thanks François Lafond, two anonymous reviewers, and the editor for useful comments on the script. This project arose as a spin-off of a wider research project on patent reclassification together with Nicolo Barbieri and Kerstin Hotte. The author thanks Nicolo and Kerstin for joint developments of the data strategy and fruitful discussions.

## Author contributions

**Conceptualization:** Peter Persoon.

**Data curation:** Peter Persoon.

**Formal analysis:** Peter Persoon.

**Methodology:** Peter Persoon.

**Validation:** Peter Persoon.

**Visualization:** Peter Persoon.

**Writing – original draft:** Peter Persoon.

**Writing – review & editing:** Peter Persoon.

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
