## [Decision Letter · Decision Letter 0]

18 Aug 2025

PONE-D-25-31819Can knowledge reclassification accelerate technological innovation?PLOS ONE

Dear Dr. Persoon,

Thank you for submitting your manuscript to PLOS ONE. After careful consideration, we feel that it has merit but does not fully meet PLOS ONE’s publication criteria as it currently stands. Therefore, we invite you to submit a revised version of the manuscript that addresses the points raised during the review process.

As you can see, the reviewers have rendered something of a split decision.  Both of them are nonetheless in agreement that this is an interesting and important topic, but have somewhat different takes on what is needed for the article to merit publication.  Despite the difference in their recommendations, I see considerable commonality in what they are asking you to do.  I encourage. you to read their comments carefully as they make a number of valuable suggestions.  Attempting to sum these up and integrate them with my own feedback, the key points that stand out for me are:

Help readers who may be less deeply familiar with the patent process to understand the phenomenon you are studying.  Perhaps provide some examples of patents that have been reclassified to make this phenomenon more concrete.  Relatedly, explain why you focus on patent families, rather than individual patents, and what the difference is both conceptually and empirically.Be clearer at the outset about what the contribution of your article is.  This is really the central concern of reviewer #2.  I think that addressing it would go some way to placating the concerns of reviewer #1 as well.  In particular are you only trying to develop a mathematical model that fits the observed patterns?  Or are you trying to explain something about the underlying processes that generate these patterns?  If the latter, then there is an inherently causal part of your story, and you need to grapple with issues of causality.  I think the reviewers read this differently and respond differently as a result.I am inclined to agree with reviewer #1 that the model terminology is difficult to follow and encourage to consider whether it can be made easier to follow, and that more intuition provided.Again, like reviewer #1 I am puzzled that you are only modeling reclassifications *into* a class.  But presumably all reclassifications must also be the result of a reclassification *out of* some other class.  Shouldn't you be modeling both of these phenomena?

In sum, I think this has the potential to be a valuable contribution, but it still requires some further work before it will merit publication.

We look forward to receiving your revised manuscript.

Kind regards,

Joshua L Rosenbloom

Academic Editor

PLOS ONE

Journal Requirements:

4. We notice that your supplementary figures are uploaded with the file type 'Figure'. Please amend the file type to 'Supporting Information'. Please ensure that each Supporting Information file has a legend listed in the manuscript after the references list.

5.  Please remove your figures from within your manuscript file, leaving only the individual TIFF/EPS image files, uploaded separately. These will be automatically included in the reviewers’ PDF.

Reviewers' comments:

Reviewer's Responses to Questions

Comments to the Author

1. Is the manuscript technically sound, and do the data support the conclusions?

Reviewer #1: No

Reviewer #2: Yes

2. Has the statistical analysis been performed appropriately and rigorously? 

Reviewer #1: No

Reviewer #2: Yes

3. Have the authors made all data underlying the findings in their manuscript fully available?

Reviewer #1: Yes

Reviewer #2: Yes

4. Is the manuscript presented in an intelligible fashion and written in standard English?

Reviewer #1: Yes

Reviewer #2: Yes

5. Review Comments to the Author

Reviewer #1: This is a sophisticated paper that takes on an important but underexplored feature of technological classification: the reclassification of patents across CPC subclasses. The author documents an interesting empirical pattern—that reclassification is pervasive, that the number of subclasses assigned to a patent increases over time, and that CPC subclasses often grow by absorbing patents from elsewhere. The paper then makes what I take to be the central modeling contribution: it distinguishes between "endemic" growth (new filings directly into a class) and growth through reclassification, and proposes that this distinction may help us better understand how different technological classes evolve.

That core insight holds promise. If some classes grow largely through direct invention and others grow by attracting work from neighboring classes, we might learn something meaningful about technological regimes, recombinative search, or the reuse of old ideas in new domains. But to make that contribution, the paper needs both a tighter modeling foundation. Below I outline my main concerns, as well as the main issues that made it hard for me to fully evaluate the paper's contribution.

1. I tried very hard to understand the model and its empirical motivation, but the use of terminology made this unnecessarily difficult. In particular, the indexing of time dimensions with t and tau (and occasionally other terms) was extremely hard to follow. I repeatedly had to go back and check whether t was filing year, observation year, or both. This could be dramatically improved by adopting more standard temporal indexing: for example, filing year or cohort year (c), and observation or calendar year (t). More generally, I think the model would be much easier to interpret if each variable had an intuitive label and was described clearly in plain language.

2. A central empirical finding of the paper is that patents are both added to and removed from classes over time. But the model only includes additions. There is no representation of reclassification out of the focal class. This is a major omission. Any generative model of class-level growth needs to account for outflows, not just inflows. As it stands, the model is not just empirically incomplete, but is logically flawed.

3. That said, the omission of reclassification-out is actually a huge opportunity. I would suggest the author reframe the paper as a decomposition exercise, and formally define three processes: alpha (endemic growth), beta (reclassification into), and omega (reclassification out of). The net change in class size over time would then be:

delta_n = alpha + beta – omega

This is a straightforward accounting identity, and it opens the door to richer empirical comparisons across classes. For example, some classes may grow mainly through alpha (internal invention), others through beta (absorbing work from elsewhere), and others may shrink through omega (having their content absorbed by other domains). These distinctions could then be linked to theoretical expectations from the innovation literature—for example, about greenfield invention vs. recombination vs. obsolescence.

4. The current write-up strongly implies causal mechanisms that are not identified. I am not a stickler on causality, but I still found the causal language to be really problematic.

5. The idea that the drop-off in recent-year patent counts is an artifact of delayed reclassification is plausible, but not appropriately tested. We would need to track cohorts of patents forward over time and show that their number of class assignments continues to grow, and that the growth is disproportionately concentrated in certain classes. Figure S3 in the supplement provides some evidence in this direction, but it's aggregate and not disaggregated by class.

6. Figure 5 is treated as empirical validation of the model in Figure 4, but it relies on the assumption that changes in patent-class assignments reflect reclassification rather than administrative changes or late additions. Without more careful documentation of the underlying data structure, this assumption is hard to verify.

7. The supplement is helpful in clarifying that most patents gain more classes than they lose. But again, this is shown in the aggregate. The model is applied at the class level, so ideally the author would provide class-level measures of inflows and outflows, and possibly their balance.

8. A very interesting interpretive possibility, not pursued in the current manuscript, is that classes with high omega values (reclassification out of the focal class) are being "liquidated" or conceptually repurposed—for example, they are no longer meaningful categories, and their content is being absorbed by newer or adjacent domains. That is an interesting phenomenon in its own right and could be explored more directly.

9. The model assumes that alpha and beta are separate processes. But the paper does not really discuss whether they are driven by different underlying mechanisms. This relates to the thinness of any theory in the paper. Could certain types of technologies be more likely to attract reclassifications? Could others be more likely to retain their boundaries?

10. The paper needs to better articulate its contribution to the innovation literature. I found the empirical pattern interesting and the proposed decomposition compelling, but I was not always sure what research question the author was trying to answer. Is the paper trying to explain technological change? The evolution of classification systems? The role of recombination? A more direct statement of the paper's aims would help a lot. As written, the paper sits between description and theory, and it's not clear what the key takeaways are meant to be. In particular, there is no theory discussed in the paper. If the author believes that reclassification is associated with growth, it would be helpful to understand the theoretical basis for that belief. Why should we expect reclassification to be positively associated with a class's growth over time?

In sum, I think the empirical pattern is real and worth studying, and the idea of decomposing class growth into endogenous invention and cross-class migration is promising. But the current model is underspecified, the write-up is hard to follow, and the framing implies causal relationships that are not supported by the analysis. I hope the author continues developing this line of work, as it has potential to contribute meaningfully to how we think about technological change and the evolution of knowledge systems.

Reviewer #2: I have one request, before moving on to some general comments. I think the model could be clearer about how one should think about positive and negative reclassifications. I have interpreted the model as describing the net effect of positive and negative reclassifications, but I would have appreciated a more explicit steer here, on page 5. I found it a bit puzzling to document positive and negative reclassifications, but then to ignore this dynamic when setting up the model. This is my only substantive request for a change, and I think it is a minor one.

The paper’s main contribution is to show that a simple mathematical equation, which describes the increase in patents assigned to a particular technology classification in a given period, is broadly consistent with actual reclassification data. A key assumption is the empirical regularity that the number of patents in a class affects the number of new patents we anticipate being assigned to the class. In particular, the more patents of a certain vintage that have been assigned to a technology class, the more likely it is that more such patents from that vintage will also be assigned to the class. But this effect dies out over time, so that reclassification gets less likely for vintages further back in time. It’s kind of a Matthew Effect combined with a decay effect.

The paper starts with these empirical regularities and demonstrates that if we assume these dynamics are the only ones governing the entire reclassification process, then we can fit a lot of features of the actual data.

I think thorny challenges remain around interpreting the significance of this result. I do not believe the paper has shown reclassification to be causal, and my null hypothesis is that the underlying dynamics are instead driven by two unobserved factors. First, new technological types arise frequently and are often a poor fit to existing classifications. These new types vary in their importance. When a new type is more important, more inventors develop inventions and seek patents for inventions of that type. More frequent contact with patent applications of the new type creates more pressure to properly assign patents of this new type to an appropriate technology class. So you get a correlation between the number of reassignments and the number of new patents sought in a class. Meanwhile, these effects might be amplified by a learning process, wherein patent examiners decide how to initially assign or later reassign a patent to a class based on the class assigned to similar patents. A patent’s most similar match is most likely to belong to a class with more rather than less patents, which creates an additional rich-get-richer effect.

I think the paper appropriately cautions against a causal interpretation of its results though, and so I think the paper overall meets the criteria prescribed by PLOS ONE for publication.

6. PLOS authors have the option to publish the peer review history of their article (what does this mean?). If published, this will include your full peer review and any attached files.

Do you want your identity to be public for this peer review? For information about this choice, including consent withdrawal, please see our Privacy Policy.

Reviewer #1: No

Reviewer #2: No

---

## [Author Response · Author response to Decision Letter 1]

5 Mar 2026

Please see a detailed overview of my reponse to each of the points raised in the 'response to reviewer' document

---

## [Decision Letter · Decision Letter 1]

16 Apr 2026

PONE-D-25-31819R1Can knowledge reclassification accelerate technological innovation?PLOS One

Dear Dr. Persoon,

Thank you for submitting your manuscript to PLOS ONE. After careful consideration, we feel that it has merit but does not fully meet PLOS ONE’s publication criteria as it currently stands. Therefore, we invite you to submit a revised version of the manuscript that addresses the points raised during the review process. Please submit your revised manuscript by May 31 2026 11:59PM. If you will need more time than this to complete your revisions, please reply to this message or contact the journal office at plosone@plos.org. Please include the following items when submitting your revised manuscript:

We look forward to receiving your revised manuscript.

Kind regards,

Joshua L Rosenbloom

Academic Editor

PLOS One

Journal Requirements:

Additional Editor Comments:

You have responded in full to the feedback from the reviewers and editor.  But there are several minor matters to attend to before proceeding to acceptance:

There is I think a probleom in equation (1) on page 6 where some symbols are not being correctly rendered.  That is I am seeing an upside down question mark and upside down exclamation mark in the first and third expressions following "if"Given the way the text is ordered it seems that the position and numbering of figures 2 and 3 might be reversed.

Please review these matters and resubmit.  I will at that point accept the paper without further review.

Reviewers' comments:

Reviewer's Responses to Questions

Comments to the Author

1. If the authors have adequately addressed your comments raised in a previous round of review and you feel that this manuscript is now acceptable for publication, you may indicate that here to bypass the “Comments to the Author” section, enter your conflict of interest statement in the “Confidential to Editor” section, and submit your "Accept" recommendation.

Reviewer #2: All comments have been addressed

2. Is the manuscript technically sound, and do the data support the conclusions?

Reviewer #2: Yes

3. Has the statistical analysis been performed appropriately and rigorously? 

Reviewer #2: Yes

4. Have the authors made all data underlying the findings in their manuscript fully available?

Reviewer #2: Yes

5. Is the manuscript presented in an intelligible fashion and written in standard English?

Reviewer #2: Yes

6. Review Comments to the Author

Reviewer #2: My original review had two main comments. First, I thought it was unclear if the model was about net reclassification or not. The revision makes this clearer, and I think that should help readers understand what the paper is doing.

Second, I thought the paper had not demonstrated causality; however, given that the paper also acknowledged this, I did not specifically request changes on this front. Nonetheless, I appreciate the many edits this revision makes to further clarify this point. I think the paper is clearer for it.

I have no further requests.

7. PLOS authors have the option to publish the peer review history of their article (what does this mean?). If published, this will include your full peer review and any attached files.

Do you want your identity to be public for this peer review? For information about this choice, including consent withdrawal, please see our Privacy Policy.

Reviewer #2: No

---

## [Author Response · Author response to Decision Letter 2]

20 Apr 2026

Dear editor and reviewers,

Thank you very much for the positive message that all reviewer points were appropriately dealt with. I hereby resubmit a version in which the remaining errors in the pdf makeup are corrected. The uploaded versions with and without annotations are identical.

Thank you very much for your effort and best wishes,

Peter Persoon

---

## [Editor Report · Decision Letter 2]

21 Apr 2026

Can knowledge reclassification accelerate technological innovation?

PONE-D-25-31819R2

Dear Dr. Persoon,

We’re pleased to inform you that your manuscript has been judged scientifically suitable for publication and will be formally accepted for publication once it meets all outstanding technical requirements.

Kind regards,

Joshua L Rosenbloom

Academic Editor

PLOS One
---

## [Editor Report · Acceptance letter]

PONE-D-25-31819R2

PLOS One

Dear Dr. Persoon,

I'm pleased to inform you that your manuscript has been deemed suitable for publication in PLOS One. Congratulations! Your manuscript is now being handed over to our production team.

Kind regards,

on behalf of

Dr. Joshua L Rosenbloom

Academic Editor

PLOS One